# Oral Semaglutide, the First Ingestible Glucagon-Like Peptide-1 Receptor Agonist: Could It Be a Magic Bullet for Type 2 Diabetes?

**DOI:** 10.3390/ijms22189936

**Published:** 2021-09-14

**Authors:** Hwi Seung Kim, Chang Hee Jung

**Affiliations:** 1Asan Medical Center, Department of Internal Medicine, University of Ulsan College of Medicine, Seoul 05505, Korea; jennyhsk212@gmail.com; 2Asan Diabetes Center, Asan Medical Center, Seoul 05505, Korea

**Keywords:** semaglutide, GLP-1 receptor, type 2 diabetes, obesity

## Abstract

The gastrointestinal tract secretes gut hormones in response to food consumption, and some of these stimulate insulin secretion. Glucagon-like peptide-1 (GLP-1) is an incretin peptide hormone released from the lower digestive tract that stimulates insulin secretion, suppresses glucagon secretion, and decreases hunger. GLP-1 receptor agonist (GLP-1RA) mimics the action of endogenous GLP-1, consequently reversing hyperglycemia and causing weight reduction, demonstrating its efficacy as an antidiabetic and antiobesity agent. Previously restricted to injection only, the invention of the absorption enhancer sodium N-(8-[2-hydroxybenzoyl]amino) caprylate resulted in the development of oral semaglutide, the first ingestible GLP-1RA. Oral semaglutide demonstrated its efficacy in glycemic management and body weight loss with a low risk of hypoglycemia as a monotherapy and in combination with other hypoglycemic medications in its clinical trial programs named Peptide Innovation for Early Diabetes Treatment. Consistent with other injectable GLP-1RAs, gastrointestinal side effects were often reported. Additionally, cardiovascular safety was established by demonstrating that oral semaglutide was not inferior to a placebo in terms of cardiovascular outcomes. Thus, oral semaglutide represents a novel treatment option that is particularly well-suited for patients with type 2 diabetes and/or obesity.

## 1. Introduction

Type 2 diabetes (T2D) is a chronic metabolic condition characterized by beta-cell dysfunction and insulin resistance that worsens over time [1]. T2D is treated with a combination of lifestyle adjustments, such as diet and exercise, as well as medication intervention [1]. Over the recent decade, new antidiabetic drugs have been introduced, expanding T2D therapy choices while also increasing treatment complexity [1,2]. Along with the development of novel agents, recent guidelines stress the patient’s coexisting diseases and risk factors before glycemic control, recommending medications with cardiovascular and renal benefits over those with glycemic control [2,3,4].

Obesity is a substantial risk factor for having T2D, and the prevalence of T2D is projected to rise as the world’s obese population grows [5]. Obesity is a modifiable risk factor, hence, correcting the etiology of obesity and insulin resistance may help prevent and treat T2D [5]. Although lifestyle changes, including diet and exercise, are essential for T2D treatment, most patients with T2D require the use of antidiabetic drugs to achieve their glycemic goals. Metformin, sulfonylureas, meglitinides, thiazolidinediones, alpha-glucosidase inhibitors, and insulin are examples of traditional agents [1]. While these medications can effectively lower blood glucose levels, many have limitations due to weight gain and hypoglycemia [1]. Furthermore, only metformin and thiazolidinediones are used to improve insulin sensitivity and affect the pathophysiology of T2D [1]. As a result, unmet needs in the treatment of T2D persist, necessitating additional research to develop new treatment choices.

### 1.1. Gut Hormones: The Metabolism Regulators

Gut hormones are involved in metabolism and interact with one another to digest the nutrients that are consumed [6] (Figure 1). Two major hormones that induce insulin production are glucose-dependent insulinotropic polypeptide (GIP) and glucagon-like peptide-1 (GLP-1) [7]. Ghrelin, on the other hand, is a stomach hormone that inhibits insulin secretion by releasing growth hormone [8]. Cholecystokinin, GLP-1, and peptide YY are hormones that slow down stomach emptying and suppress appetite [9,10]. Glucagon is produced in the pancreatic alpha cells and increases hepatic glucose synthesis and lipolysis to boost blood glucose levels [11]. Pancreatic polypeptide excreted from the pancreas is involved in the long-term regulation of appetite [10]. Oxyntomodulin, released concurrently with GLP-1 by the L-cells, reduces food intake while increasing energy consumption and insulin secretion [12,13]. The aforementioned gut hormones are being studied in order to find new potential medications to address the unmet demand in T2D treatment.

### 1.2. Glucagon-like Peptide-1 (GLP-1): An Innovator for Gut Hormone Therapies

The release of incretin hormones after eating a meal has opened new avenues for the development of antidiabetic drugs [14]. Incretins promote insulin release from pancreatic beta cells in response to hyperglycemia, preserving normoglycemia [15]. Both GIP and GLP-1, which are each released by K-cells in the duodenum and upper jejunum and by L-cells in the distal ileum and large intestines, enhance glucose-mediated insulin release [16,17]. In patients with T2D, however, GIP loses much of its insulinotropic activity, whereas GLP-1 demonstrates a sustained but diminished insulinotropic response [18]. Furthermore, while GLP-1 reduces glucagon secretion in a glucose-dependent manner, GIP has no effect on glucagon secretion during hyperglycemia and instead increases it during hypoglycemia [19]. GLP-1 has also been demonstrated to have pleiotropic effects, such as lowering hunger and food intake, as well as slowing stomach emptying and small bowel movement [20,21]. As a result, GLP-1 has become a hot target for possible T2D and obesity medicines.

GLP-1 receptor agonist (GLP-1RA) decreases food intake by increasing gastric emptying time and satiety, resulting in body weight loss [22]. Despite its antidiabetic and antiobesity properties, the GLP-1RA has been restricted in usage because of gastrointestinal side effects (nausea, vomiting, constipation, and abdominal discomfort) [7]. Furthermore, despite its established efficacy, the subcutaneous injection technique of administering GLP-1RA has further limited its prescription. Concerns with injection, including pain and fear, have been noted in studies as barriers to maintaining GLP-1RA in real-world practice [23,24]. As a result, the first oral GLP-1RA semaglutide is a significant achievement, offering a practical alternative to injectables for patients with T2D.

The goal of this review was to describe the pharmacology, efficacy, safety, and clinical implications of oral semaglutide, the first ingestible gut hormone derivative, as a T2D treatment.

## 2. Pharmacodynamics and Pharmacokinetics of Oral Semaglutide

Semaglutide was originally designed as a once-weekly subcutaneous long-acting GLP-1RA. Semaglutide is a human GLP-1 analog with 94% similarity to natural human GLP-1 but has amino acid changes that improve albumin binding, decrease renal clearance, and boost resistance to DPP-4 destruction [25]. Semaglutide demonstrated efficacy in glycemic control and body weight reduction compared to placebo and active comparators, such as sitagliptin, exenatide extended-release, dulaglutide, and insulin glargine, in the Semaglutide Unabated Sustainability in Treatment of Type 2 Diabetes (SUSTAIN) clinical trials [26,27,28,29,30]. Furthermore, semaglutide improved cardiovascular outcomes significantly [31].

As the peptide-based drug is degraded by proteolytic enzymes and the pH of the gastrointestinal tract, semaglutide must be injected subcutaneously. Semaglutide could be developed as an oral tablet by combining it with an absorption enhancer known as sodium N-(8-[2-hydroxybenzoyl]amino) caprylate (SNAC) [32,33] (Figure 2). In a concentration-dependent manner, SNAC forms a noncovalent bond with GLP-1, increasing lipophilicity and transcellular absorption of semaglutide through the stomach epithelium [32,34]. Additionally, in the acidic environment of the stomach, SNAC acts as a local pH buffer for semaglutide, increasing solubility and protecting the drug from degradation [34]. As SNAC’s activity is brief and reversible, it separates from the medication once it reaches the bloodstream [32].

In contrast to most other medications absorbed in the intestines, oral semaglutide has a distinct pharmacokinetic profile since it is virtually fully absorbed in the stomach. After 15–35 min of oral intake, the medication reaches its maximal concentration [32]. As food interferes with drug absorption, oral semaglutide should be given while fasting [35]. Semaglutide exposure was unaffected by the amount of water used to take the drug [35]. However, the higher the semaglutide exposure, the longer the post-dose fasting time; hence, at least 30 min of post-dose fasting is recommended [35]. Renal and hepatic impairment had no effect on the pharmacokinetics of oral semaglutide, indicating that individuals with renal and hepatic impairment do not require dose adjustments [36,37]. In investigations involving omeprazole, lisinopril, warfarin, digoxin, metformin, levonorgestrel, ethinyl estradiol, furosemide, and rosuvastatin, no significant drug–drug interactions were found [38,39,40]. Thyroid function tests should be monitored in patients receiving both oral semaglutide and levothyroxine since the pharmacokinetics of levothyroxine are influenced by a 33% increase in exposure when taken with oral semaglutide [41].

## 3. Clinical Efficacy and Safety: Summary of Peptide Innovation for Early Diabetes Treatment (PIONEER) Trials

The PIONEER program was composed of phase 3 clinical trials to compare the clinical efficacy and safety of oral semaglutide against placebo and other medications. The program included 10 trials, eight of which were global and two of which were solely undertaken in Japan. The PIONEER studies, which began in 2016 and ended in 2018, enrolled 9543 participants (1293 Japanese patients). The PIONEER program used two estimands established by the International Council of Harmonisation to evaluate the efficacy objectives: “treatment policy” and “trial product” [42]. The treatment policy estimand is based on the intention-to-treat principle and covers all randomized patients, even those who stopped taking the test medicine and added or switched to something else. The trial product estimand, on the other hand, assesses the therapeutic efficacy of individuals who stayed on the test drug without receiving any rescue medication. This review focuses mostly on the treatment policy estimand. The results of the PIONEER program regarding HbA1c and weight reduction are each presented in Table 1 and Table 2.

### 3.1. Placebo-Controlled Trials

#### 3.1.1. Monotherapy

The PIONEER 1 trial evaluated the efficacy and safety of oral semaglutide as a monotherapy in patients with T2D who were not taking any antidiabetic drugs [43]. In a 1:1:1:1 ratio, 703 participants were randomly assigned to receive 3, 7, or 14 mg of oral semaglutide or placebo [43]. Participants who were given oral semaglutide began with 3 mg and the dose was gradually raised every 4 weeks until the goal dose was reached [43]. The participants were 55 years old on average, 50.8% were male, and all had diabetes for mean 3.5 years [43]. At baseline, the mean HbA1c was 8.0% and the mean BMI was 31.8 kg/m^2^ [43]. When compared to the placebo, oral semaglutide resulted in a larger reduction in HbA1c, with placebo-adjusted differences of −0.6%, −0.9%, and −1.1% for 3, 7, and 14 mg, respectively [43]. The proportions of individuals attaining the target HbA1c of <7% and ≤6.5% were significantly higher in the oral semaglutide group compared to the placebo group (*p* < 0.001) [43]. When compared to the placebo, the 14 mg oral semaglutide demonstrated greater weight loss [43]. The placebo-adjusted differences in body weight were −0.1 (*p* = 0.87), −0.9 (*p* = 0.09), and −2.3 kg (*p* < 0.001) for 3, 7, and 14 mg, respectively [43]. Oral semaglutide was found to be more efficacious than placebo as a monotherapy in patients with T2D in the PIONEER 1 study, and the side effects were similar to those seen with other injectable GLP-1RAs [43]. Dose-dependent HbA1c and weight reduction was observed in oral semaglutide versus placebo at higher dosages.

#### 3.1.2. Combination Therapy

The PIONEER 8 trial enrolled 731 individuals who were using either basal insulin or metformin [44]. Patients with a mean age of 61 years and a diabetes duration of 15 years were randomized to receive oral semaglutide 3, 7, or 14 mg or placebo [44]. Patients were given a 20% lower insulin dose before starting the study drug, which they continued for 8 weeks until an increase was required [44]. The insulin dose could be modified within the pre-randomization dose during weeks 8–26, and the insulin dose could be freely adjusted during weeks 26–52 [44]. In comparison with the placebo, the estimated treatment difference (ETD) in the mean change in HbA1c was −0.4% (*p* = 0.0004), −0.6% (*p* = 0.0001), and −0.9% (*p* = 0.0001) for 3, 7, and 14 mg of oral semaglutide, respectively [44]. At week 52, all patients taking semaglutide doses lost considerably more weight than those taking the placebo, with ETDs of −0.8 (*p* = 0.0101), −2.5 (*p* < 0.0001), and −4.3 kg (*p* < 0.0001) for 3, 7, and 14 mg of oral semaglutide, respectively [44]. At 52 weeks, oral semaglutide resulted in a statistically significant decrease in total daily insulin dose; in comparison to the placebo, semaglutide 3 mg had −8 units (*p* = 0.0450), semaglutide 7 mg had −16 units (*p* < 0.0001), and semaglutide 14 mg had −17 units (*p* < 0.0001) [44]. Nausea was the most commonly reported side effect in the semaglutide group (11.4–23.2%) compared to 7.1% in the placebo group [44]. The PIONEER 8 trial found that oral semaglutide, when combined with insulin and metformin, was effective and safe [44]. Oral semaglutide 7 and 14 mg provided better glycemic control than placebo with lower total daily insulin dosages at both weeks 26 and 52, supporting the use of oral semaglutide to intensify treatment for individuals unable to achieve glycemic target with insulin alone.

### 3.2. Active-Comparator Trials

#### 3.2.1. Sodium Glucose Co-Transporter 2 (SGLT-2) Inhibitors

In the PIONEER 2 trial, patients with T2D on metformin monotherapy were compared to those on oral semaglutide and empagliflozin [45]. Oral semaglutide 14 mg and empagliflozin 25 mg were randomly allocated to 412 and 410 patients [45]. The patients were on average 58 years old, 49.5% were female, and the mean body weight was 91.6 kg [45]. Oral semaglutide lowered HbA1c by 1.3% and empagliflozin by 0.9% after 52 weeks (*p* < 0.0001) [45]. In comparison to empagliflozin, the proportion of patients achieving HbA1c 7% was also significantly greater in the oral semaglutide groups (66.1% vs. 43.2%, *p* < 0.0001) [45]. Body weight change was −3.8 kg in semaglutide and −3.6 kg in empagliflozin (*p* = 0.6231); however, the trial product estimand at week 52 showed that oral semaglutide reduced body weight significantly more than empagliflozin (−4.7 versus −3.8 kg; *p* = 0.0114) [45]. Adverse events, mostly mild to moderate in severity, were reported in 70.5% and 69.2% of patients on oral semaglutide and empagliflozin, respectively [45]. Nausea was the most common side effect in the semaglutide group, with 19.8% experiencing it [45]. Oral semaglutide was found to be superior to empagliflozin in lowering HbA1c and weight (trial product estimand) at week 52 in the PIONEER 2 trial, and to be as tolerable as injectable GLP-1RAs [45]. In patients with T2D who were uncontrolled on metformin alone, oral semaglutide significantly improved HbA1c versus empagliflozin, while fasting glucose reductions were similar in both groups, implying that changes in glycemic control are mostly due to the greater reduction of postprandial glucose with oral semaglutide.

#### 3.2.2. DPP-4 Inhibitors

In the PIONEER 3 trial, 1864 patients with T2D who were using metformin with or without sulfonylurea were compared to those using oral semaglutide versus sitagliptin [46]. For 78 weeks, patients were randomly assigned 1:1:1:1 to oral semaglutide 3, 7, or 14 mg or sitagliptin 100 mg [46]. The semaglutide dose was started at 3 mg and gradually increased to the randomized dose every 4 weeks. The average age was 58 years old and the average BMI was 32.5 kg/m^2^ [46]. Compared to −0.8% for sitagliptin, the estimated mean changes in HbA1c at week 26 for 3, 7, and 14 mg of semaglutide were −0.6%, −1.0%, and −1.3%, respectively [46]. Semaglutide 7 and 14 mg both outperformed sitagliptin in terms of glucose control (*p* < 0.001); however, semaglutide 3 mg failed to exhibit non-inferiority to sitagliptin [43]. Semaglutide 3, 7, and 14 mg reduced HbA1c by 0.0% (*p* = 0.61), 0.1% (*p* = 0.06), and 0.4% (*p* < 0.001), respectively, at week 78 [46]. In terms of weight loss at week 78, dosages of 3, 7, and 14 mg of oral semaglutide outperformed sitagliptin by 0.8, 1.7, and 2.1 kg (*p* = 0.02 for 3 mg and *p* < 0.001 for 7 and 14 mg), respectively [46]. The authors found that oral semaglutide dosages of 7 and 14 mg were superior to sitagliptin in terms of glycemic management and weight loss, and the safety profile was comparable to that of other injectable GLP-1RAs [46]. As enhanced glycemic control is linked to better diabetes-related outcomes and some patients prefer oral drugs, the results obtained with oral semaglutide may be clinically relevant.

In the PIONEER 7 trial, sitagliptin was again compared to oral semaglutide, this time with flexibly dosed oral semaglutide [47]. A total of 504 patients with uncontrolled T2D who were using one or both oral hypoglycemic drugs were randomly assigned to receive either oral semaglutide or sitagliptin [47]. Patients who were randomly assigned to the oral semaglutide group were given 3 mg for 8 weeks and subsequently had their doses changed every 8 weeks based on their HbA1c level and tolerability [47]. The participants were 57% male and the mean age was 57.4 years [47]. At week 8, 73% of participants in the semaglutide group were able to receive an increased dose of 7 mg [47]. At week 52, 9%, 30%, and 59% of patients received oral semaglutide dosages of 3, 7, and 14 mg, respectively [47]. Semaglutide was significantly greater than sitagliptin at lowering the HbA1c and weight with an ETD of −0.5% (*p* < 0.0001) and −1.9 kg (*p* < 0.0001) [47]. There were many adverse events reported, including 78% in the oral semaglutide group and 69% in the sitagliptin group, and nausea was the most common [47]. In the PIONEER 7 trial, flexibly dosed oral semaglutide outperformed sitagliptin in lowering HbA1c and body weight [47]. However, contrary to the expectations, flexible dose adjustment did not reduce the prevalence of adverse events.

#### 3.2.3. Injectable GLP-1RAs

The PIONEER 4 trial compared the efficacy of oral semaglutide versus liraglutide in 711 patients with T2D during a 52 week period [48]. Patients on metformin with or without an SGLT-2 inhibitor who had a HbA1c of 7.0–9.5% were randomly assigned 2:2:1 to oral semaglutide, subcutaneous liraglutide, or placebo [48]. Semaglutide and liraglutide doses were gradually increased to 14 and 1.8 mg, respectively [48]. At week 26, oral semaglutide lowered HbA1c by 1.2%, while liraglutide and placebo showed 1.1% and 0.2% HbA1c reduction, respectively (*p* = 0.0645 between semaglutide and liraglutide, *p* < 0.0001 between semaglutide and placebo) [48]. Oral semaglutide, on the other hand, demonstrated a greater reduction of HbA1c compared to both liraglutide and placebo at week 52, with ETDs of −0.3% compared to liraglutide (*p* = 0.0002) and −1.0% compared to placebo (*p* < 0.0001) [48]. Additionally, oral semaglutide reduced the body weight considerably more than both liraglutide and placebo at week 26 (ETD −1.2 kg, *p* = 0.0003 for liraglutide and ETD −3.8 kg, *p* < 0.0001 for placebo) and week 52 (ETD −1.3 kg, *p* = 0.0019 for liraglutide and ETD −3.3 kg, *p* < 0.0001 for placebo) [48]. Oral semaglutide is non-inferior to subcutaneous liraglutide and superior to placebo in HbA1c reduction and greater than liraglutide and placebo in weight loss; thus, oral semaglutide may be a better option for patients who refuse injectable GLP-1RA [48]. Being the first to compare oral versus subcutaneous GLP-1RAs for T2D, this study suggested the long-term (52-week) efficacy and safety of oral semaglutide versus subcutaneous liraglutide.

The PIONEER 9 trial, which was conducted in Japan, compared the efficacy of oral semaglutide monotherapy with that of placebo and liraglutide in 243 Japanese patients with T2D who were inadequately controlled with diet and exercise alone or one oral hypoglycemic agent that had to be washed out before starting the study drug [49]. Patients were randomly assigned to receive either 3, 7, or 14 mg of oral semaglutide, once-daily subcutaneous liraglutide 0.9 mg (the maximum dose allowed in Japan), or placebo [49]. Liraglutide was started at 0.3 mg per day through subcutaneous injection and increased by 0.3 mg every week until it reached 0.9 mg [49]. After 26 weeks, all of the three oral semaglutide groups had considerably lower A1c levels than placebo (*p* < 0.0001), and the semaglutide 14 mg group had significantly lower A1c levels than the liraglutide 0.9 mg group (*p* = 0.0272) [49]. At 52 weeks, all three dosages of oral semaglutide remained superior to placebo (*p* < 0.0001), whereas the difference between oral semaglutide and liraglutide was not significant [49]. In terms of HbA1c lowering, oral semaglutide 7 mg was comparable to liraglutide 0.9 mg, and oral semaglutide 14 mg was superior to liraglutide 0.9 mg in Japanese patients with T2D.

In PIONEER 10, oral semaglutide and weekly dulaglutide were compared in Japanese patients with T2D [50]. A total of 458 participants were enrolled in the study and were randomly assigned to receive semaglutide 3, 7, or 14 mg, or 0.75 mg of dulaglutide [50]. By week 52, changes in HbA1c from baseline with oral semaglutide 3, 7, and 14 mg were −0.9%, −1.4%, and −1.7%, respectively, compared to −1.4% with dulaglutide [50]. Only the 14 mg dose of oral semaglutide was significantly superior to dulaglutide with an ETD of −0.3% (*p* = 0.0170) [50]. With ETDs of −1.9 and −2.6 kg, oral semaglutide 7 and 14 mg showed considerable superiority to dulaglutide in terms of body weight change (*p* < 0.0001 for both) [50]. Adverse events were reported in 77% (101/131) of the oral semaglutide 3 mg group, 80% (106/132) of the oral semaglutide 7 mg group, 85% (111/130) of the oral semaglutide 14 mg group, and 82% (52/65) of the dulaglutide group, indicating a safety profile similar to that of other GLP-1RAs [50]. The fixed dosage of dulaglutide in this experiment is a limitation, as dulaglutide can be titrated up to 1.5 mg [50]. The authors concluded that oral semaglutide 14 mg was superior to dulaglutide 0.75 mg in terms of HbA1c reduction and semaglutide 7 and 14 mg in terms of weight reduction [50]. Once-daily oral semaglutide was also well-tolerated, with the incidence of adverse events similar to once-weekly subcutaneous dulaglutide.

## 4. Special Populations

In patients with T2D and renal impairment, the PIONEER 5 study compared the effectiveness and tolerability of oral semaglutide versus placebo [51]. Participants were on a stable dose of metformin and/or sulfonylurea or basal insulin with or without metformin for at least 90 days with an estimated glomerular filtration rate of 30–59 mL/min/1.73 m^2^ [51]. In addition to their current medication, 324 individuals were randomly allocated to receive oral semaglutide (dosage gradually increased from 3 to 14 mg) or placebo for 26 weeks [51]. HbA1c and body weight reduction showed superiority in the oral semaglutide group compared to the placebo group with an ETD of −0.8% of HbA1c (*p* < 0.0001) and −2.5 kg of body weight (*p* < 0.0001) [51]. In the oral semaglutide group, the urine albumin to creatinine ratio improved, but it worsened in the placebo group [51]. From baseline to week 26, mean systolic and diastolic blood pressures (BPs) were reduced by 7 and 2 mmHg, respectively, with statistically significant ETDs compared to placebo (−7 mmHg, *p* < 0.0001 for systolic BP and −3 mmHg, *p* = 0.0018 for diastolic BP) [51]. Adverse events were consistent with those of prior studies, and renal function did not alter in either treatment group during the trial [51]. Overall, oral semaglutide was efficacious and safe in individuals with renal dysfunction, demonstrating its potential as an antidiabetic medication that may improve kidney function in patients with T2D with decreased renal function [51].

## 5. Cardiovascular Outcomes with Oral Semaglutide

The goal of the PIONEER 6 trial was to assess the cardiovascular safety of oral semaglutide [52]. In total, 3183 patients with T2D who were receiving conventional therapy were enrolled and randomly assigned to receive either oral semaglutide or a placebo [52]. All eligible patients were at high risk of cardiovascular disease, either ≥ 50 years with established cardiovascular or renal disease or ≥ 60 years with cardiovascular risk factors [52]. A cardiovascular event, such as cardiovascular death, non-fatal myocardial infarction, or non-fatal stroke, was the primary outcome [52].

During the median follow-up of 15.9 months, 137 major adverse cardiovascular events (MACEs) (61/1591 patients (3.8%) in the oral semaglutide group and 76/1592 patients (4.8%) in the placebo group) were documented [52]. Oral semaglutide was found to be non-inferior to placebo in terms of cardiovascular risk, with a difference of 21% (hazard ratio (HR) 0.79; *p* < 0.001). However, the HR was not significant for superiority (*p* = 0.17) [52]. Nonetheless, when compared to the placebo group, death from cardiovascular causes and death from any cause were significantly lower in the oral semaglutide group (HR 0.49, 95% confidence interval (CI) 0.27–0.92 and HR 0.51, 95% CI 0.31–0.84, respectively) [50]. Furthermore, compared to 0.3% and 0.8 kg in the placebo group, the oral semaglutide group had a 1.0% drop in HbA1c and 4.2 kg weight loss [52]. As a result, the authors determined that oral semaglutide was non-inferior to placebo in terms of cardiovascular outcomes [52].

Patients in the PIONEER 6 trial had the same cardiovascular risk profile as those in the SUSTAIN 6 trial [30]. As a result, the composite primary outcome was as well-established as the first occurrence of a MACE (death from cardiovascular causes, non-fatal myocardial infarction, and non-fatal stroke). However, whereas the SUSTAIN 6 study demonstrated that subcutaneous semaglutide is superior to placebo (HR 0.74; 95% CI 0.58–0.95), the PIONEER 6 trial could only demonstrate that oral semaglutide was non-inferior to placebo (HR 0.79; 95% CI 0.57–1.1) [30,52]. Given that the patients’ baseline characteristics were similar in both studies, the disparity in cardiovascular outcomes could have resulted from the difference in the number of cardiovascular events due to the difference in trial duration between the PIONEER 6 (137 events in 64 weeks) and SUSTAIN 6 (254 events in 104 weeks) trials [30,52]. However, the varied form of administration could be a factor for the discrepancy in cardiovascular outcomes; hence, further studies comparing oral and subcutaneous semaglutide would be beneficial.

## 6. Future Perspectives

The phase 3 randomized clinical trials PIONEER 11 and 12 are now underway in China and are expected to be finished this year. The PIONEER 11 trial compares oral semaglutide to placebo in patients with T2D who are not receiving any antidiabetic drugs. The primary outcome is the change in HbA1c from baseline to week 26 [53]. The PIONEER 12 trial evaluates the efficacy and safety of oral semaglutide compared to sitagliptin in patients showing HbA1c of 7–10.5% on a stable dose of metformin [54]. Again, the primary outcome is the change in HbA1c from baseline to week 26 [54]. These two clinical trials will contribute to the evidence of oral semaglutide’s efficacy and tolerability.

In the PIONEER 6 trial, oral semaglutide only revealed non-inferiority to placebo in terms of cardiovascular benefits; as a result, a longer-term investigation, A Heart Disease Study of Semaglutide in Patients with Type 2 Diabetes (SOUL), is currently underway [55]. SOUL has enrolled around 9640 patients with T2D and cardiovascular disease, cerebrovascular disease, symptomatic peripheral artery disease, or chronic kidney disease for 3.5–5 years [55]. In addition, the time to a MACE occurrence in patients randomized to oral semaglutide 14 mg or placebo is being studied [55].

## 7. Conclusions

Oral semaglutide, which was approved in the United States in September 2019, has emerged as a promising therapeutic option for patients with T2D [56]. Oral semaglutide 14 mg lowered HbA1c substantially more than placebo, empagliflozin, sitagliptin, liraglutide, and dulaglutide in the PIONEER clinical study program. The oral semaglutide 14 mg group lost considerably more weight than the placebo, sitagliptin, and liraglutide groups. Furthermore, oral semaglutide was well-tolerated, with an adverse event profile similar to that of subcutaneous GLP-1RAs. In the PIONEER 6 trial, cardiovascular safety was also demonstrated. As a result, oral semaglutide has been identified as a convenient and successful therapy option for patients with T2D, allowing for earlier administration of GLP-1RA.

Gut hormones, which play a key role in regulating energy balance and metabolism, are being touted as potential new antidiabetic and antiobesity therapeutic targets. Agents that interact with two or three gut hormone receptors have been demonstrated to be effective in improving insulin sensitivity and weight reduction [7,57]. Following the successful launch of the absorption enhancer in oral semaglutide, attempts to experiment with and develop hormone-based therapies will only increase. Novel medications based on gut hormones promise a bright future for the treatment of diabetes and obesity.

## Figures and Tables

**Figure 1 ijms-22-09936-f001:**
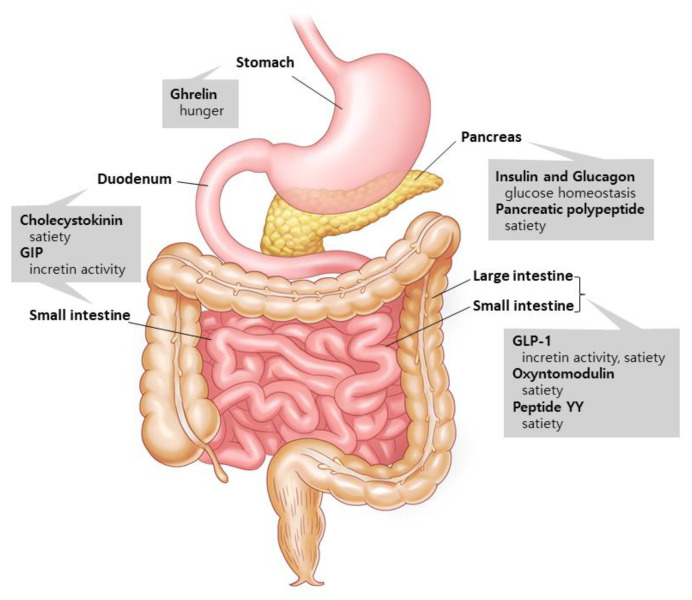
Gut hormones and their supposed actions. Ghrelin is released by the stomach. Insulin, glucagon, and pancreatic polypeptide are excreted from the pancreas. Cholecystokinin and GIP are secreted in the duodenum and small intestines, and GLP-1, oxyntomodulin, and peptide YY are released by the small and large intestines. These hormones from the gastrointestinal tract communicate with the peripheral and central nervous systems to control a variety of metabolic activities. Modified from Murphy et al. [6].

**Figure 2 ijms-22-09936-f002:**
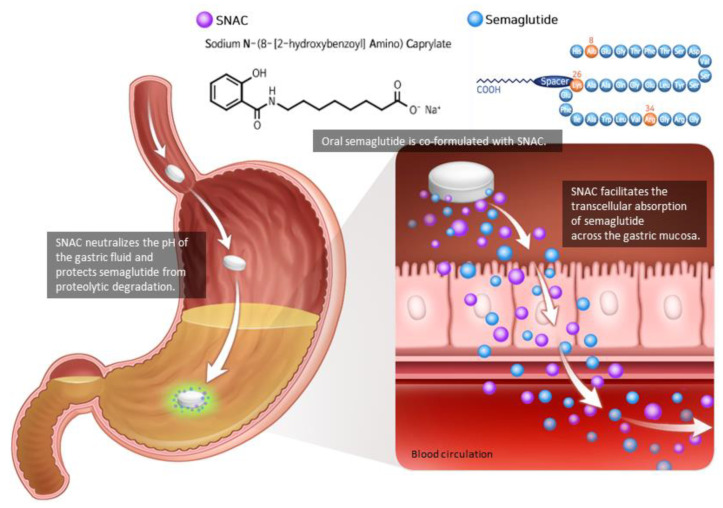
Oral semaglutide and SNAC. Oral semaglutide must be co-formulated with the absorption enhancer SNAC in order to be absorbed. SNAC raises the local pH, resulting in increased solubility and protection from proteolytic degradation. SNAC promotes the absorption of semaglutide across the gastric mucosa in a time- and concentration-dependent manner, which is totally reversible. Modified from Andersen et al. [33].

**Table 1 ijms-22-09936-t001:** Summary of efficacy in glycemic control across the PIONEER trials [43,44,45,46,47,48,49,50,51].

	Time (wk)	No. of Patients (Japanese)	Comparator	Baseline HbA1c (%)	Mean Reduction in HbA1c (%)			
					Semaglutide			Comparator
					3 mg	7 mg	14 mg	
PIONEER 1	26	703 (116)	Placebo	8.0	−0.9 *	−1.2 *	−1.4 *	−0.3
PIONEER 2	52	822 (0)	Empagliflozin 25 mg	8.1			−1.3 *	−0.9
PIONEER 3	78	1864 (207)	Sitagliptin 100 mg	8.3	−0.6	−1.0 *	−1.3 *	−0.8
PIONEER 4	52	711 (75)	Liraglutide 1.8 mg or placebo	8.0			−1.2	−1.1 (liraglutide) −0.2 (placebo)
PIONEER 5	26	324 (0)	Placebo (renal)	8.0			−1.0 *	−0.2
PIONEER 7	52	504 (0)	Sitagliptin 100 mg	8.3	−1.3 * (flexible dosing)			−0.8
PIONEER 8	52	731 (194)	Placebo (add-on to insulin)	8.2	−0.6 *	−0.9 *	−1.3 *	−0.1
PIONEER 9	52	243 (243)	Liraglutide 0.9 mg or placebo	8.2	−0.9 ^¶^	−1.4 ^¶^	−1.5 ^¶^	−1.2 (liraglutide) −0.1 (placebo)
PIONEER 10	52	458 (458)	Dulaglutide 0.75 mg	8.3	−0.9 *	−1.4	−1.7 *	−1.4

* Statistically significant compared to the comparator. ^¶^ Statistically significant compared to the placebo.

**Table 2 ijms-22-09936-t002:** Summary of efficacy in weight reduction across the PIONEER trials [43,44,45,46,47,48,49,50,51].

	Time (wk)	No. of Patients (Japanese)	Comparator	Baseline Weight (kg)	Mean Reduction in Weight (kg)			
					Semaglutide			Comparator
					3 mg	7 mg	14 mg	
PIONEER 1	26	703 (116)	Placebo	88.1	−1.5	−2.3	−3.7 *	−1.4
PIONEER 2	52	822 (0)	Empagliflozin 25 mg	91.6			−3.8	−3.7
PIONEER 3	78	1864 (207)	Sitagliptin 100 mg	91.2	−1.2 *	−2.2 *	−3.1 *	−0.6
PIONEER 4	52	711 (75)	Liraglutide 1.8 mg or placebo	94.0			−4.4 *	−3.1 (liraglutide) −0.2 (placebo)
PIONEER 5	26	324 (0)	Placebo (renal)	90.8			−3.4 *	−0.9
PIONEER 7	52	504 (0)	Sitagliptin 100 mg	88.6	−2.6 * (flexible dosing)			−0.7
PIONEER 8	52	731 (194)	Placebo (add-on to insulin)	85.9	−1.4 *	−2.4 *	−3.7 *	−0.4
PIONEER 9	52	243 (243)	Liraglutide 0.9 mg or placebo	71.1	−0.3	−0.8	−2.6 ^†,¶^	−0.6 (placebo) 0.0 (liraglutide)
PIONEER 10	52	458 (458)	Dulaglutide 0.75 mg	72.1	−0.0 *	−0.9 *	−1.6 *	1.0

* Statistically significant compared to the comparator. ^†^ Statistically significant compared to liraglutide. ^¶^ Statistically significant compared to the placebo.

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
