# Peer review of "Oral Semaglutide, the First Ingestible Glucagon-Like Peptide-1 Receptor Agonist: Could It Be a Magic Bullet for Type 2 Diabetes?"

_ijms, 2021, doi:10.3390/ijms22189936_

Round 1

Reviewer 1 Report

Review of the manuscript entitled: “Oral Semaglutide, the First Ingestible Glucagon-Like Peptide-1 Receptor Agonist: Could it be a Magic Bullet for Type 2 Diabetes?”. Manuscript is prepared very well but there are a few comments that should be taken into account. In my opinion chapters 1, 2 and 3 are an introduction in lines 92-94 is aim of review. Maybe it would be better to do everything as one chapter “Introduction”? Or  subheadings for example 1.1. Gut Hormones: The Metabolism Regulators, 1.2. Glucagon-Like Peptide-1 (GLP-1): An Innovator for Gut Hormone Therapies. Tables 1 and 2 should be centered. Line 166 – something missing.

Author Response

Thank you for the constructive comments. We agree with your opinion about chapters 1, 2, and 3; therefore, we put chapters 2 and 3 as subheadings 1.1 and 1.2 under Chapter 1 Introduction. In addition, Tables 1 and 2 were formatted to be centered. Line 166 was cut off because of Table 1, so we revised the format to correct the mistakes in formatting.

Reviewer 2 Report

This review examines the efficacy and implications for treatment using semaglutide as a GLP1 receptor agonist when orally administered by combing with the absorption enhancer SNAC.  The article is generally well-written and accessible to a wide audience. It provides a simplistic overview for gut hormones and available treatments for T2DM before presenting available data on phacodynamics/kinetics, efficacy within monotherapy and polypharmacy settings and potential CV contraindications compared to injectable formats of the drug and other treatment options, e.g., subcutaneous liraglutide. There is an appreciation for the lack of drug-drug interactions and a summary of efficacy and safety data from the PIONEER trails with a focus on tolerability, improved HbA1c and weight.

As a review for the efficacy in glycaemic control reported from the PIONEER trials the report provides a snapshot of benefits and CV safety of oral semaglutide and I have very little criticism other than a request that Table 1 be positioned so that the whole table can be accessed on the same page.  Overall, the work is a summary of the data collected in the PIONEER trials and the article might be improved by giving more consideration to discussing the findings rather than merely reiterating earlier findings.  However, the authors do make the data accessible and I can understand the merit of breaking things down into a digestible format.

Author Response

Thank you for the valuable comments. We changed the format of Table 1 so that it is centered.  Following your suggestion about discussing the findings rather than merely reiterating them, we additionally discussed the findings of the PIONEER trials at the end of each paragraph under title 3. Clinical Efficacy and Safety: Summary of PIONEER trials. The added sentences are provided below.

3.1.1. Monotherapy

“Dose-dependent HbA1c and weight reduction was observed in oral semaglutide versus placebo at higher dosages.” (page 5, lines 169-171).

3.1.2 Combination therapy

“Oral semaglutide 7 and 14 mg provided better glycemic control than placebo with lower total daily insulin dosages at both week 26 and 52, supporting the use of oral semaglutide to intensify treatment for individuals unable to achieve glycemic target with insulin alone.” (page 6, lines 199-202).

3.2.1 Sodium glucose co-transporter 2 (SGLT-2) inhibitors

“In patients with T2D who were uncontrolled on metformin alone, oral semaglutide significantly improved HbA1c versus empagliflozin, while fasting glucose reductions were similar in both groups, implying that changes in glycemic control are mostly due to the great-er reduction of postprandial glucose with oral semaglutide.” (page 6, lines 220-224)

3.2.2 DPP-4 inhibitors

“Because enhanced glycemic control is linked to better diabetes-related outcomes and some patients prefer oral drugs, the results obtained with oral semaglutide may be clinically relevant.” (page 7, lines 241-243).

“However, contrary to the expectations, flexible dose adjustment did not reduce the prevalence of adverse events.” (page 7, lines 264-265).

3.2.3 Injectable GLP-1RAs

“Being the first to compare oral versus subcutaneous GLP-1RAs for T2D, this study suggested the long-term (52-week) efficacy and safety of oral semaglutide versus subcutaneous liraglutide.” (page 8, lines 283-285).

“In terms of HbA1c lowering, oral semaglutide 7 mg was comparable to liraglutide 0.9 mg, and oral semaglutide 14 mg was superior to liraglutide 0.9 mg in Japanese patients with T2D.” (page 8, lines 298-300).

“Once-daily oral semaglutide was also well-tolerated, with the incidence of adverse events similar to once-weekly subcutaneous dulaglutide.” (page 8, lines 316-317).